# When Size *Really* Matters: The Eccentricities of Dystrophin Transcription and the Hazards of Quantifying mRNA from Very Long Genes

**DOI:** 10.3390/biomedicines11072082

**Published:** 2023-07-24

**Authors:** John C. W. Hildyard, Richard J. Piercy

**Affiliations:** Comparative Neuromuscular Disease Laboratory, Department of Clinical Science and Services, Royal Veterinary College, London NW1 0TU, UK; rpiercy@rvc.ac.uk

**Keywords:** DMD, dystrophin, gene expression, transcription, mRNA, RNAseq

## Abstract

At 2.3 megabases in length, the dystrophin gene is enormous: transcription of a single mRNA requires approximately 16 h. Principally expressed in skeletal muscle, the dystrophin protein product protects the muscle sarcolemma against contraction-induced injury, and dystrophin deficiency results in the fatal muscle-wasting disease, Duchenne muscular dystrophy. This gene is thus of key clinical interest, and therapeutic strategies aimed at eliciting dystrophin restoration require quantitative analysis of its expression. Approaches for quantifying dystrophin at the protein level are well-established, however study at the mRNA level warrants closer scrutiny: measured expression values differ in a sequence-dependent fashion, with significant consequences for data interpretation. In this manuscript, we discuss these nuances of expression and present evidence to support a transcriptional model whereby the long transcription time is coupled to a short mature mRNA half-life, with dystrophin transcripts being predominantly nascent as a consequence. We explore the effects of such a model on cellular transcriptional dynamics and then discuss key implications for the study of dystrophin gene expression, focusing on both conventional (qPCR) and next-gen (RNAseq) approaches.

## 1. Introduction

### 1.1. The Dystrophin Gene

The dystrophin gene is one of the largest in the mammalian genome: spanning approximately 2.3 megabases of the X chromosome, this single gene accounts for almost one-thousandth of total genomic DNA (Figure 1A). Much of this sequence is noncoding: the majority of the 79 exons of the canonical full-length dystrophin mRNA are shorter than 150 bases, typically interspersed with introns that are thousands (or even hundreds of thousands) of bases long, all of which are transcribed in full. Accordingly, the transcription of a single full-length mRNA requires approximately 16 h, and splicing occurs co-transcriptionally [1], with some long introns spliced in a sequential, multistage process [2]. Mature dystrophin mRNA is ~14 kb in length (Figure 1B,C), less than 1% of the size of the dystrophin gene, yet still a large transcript (even the 2.5 kb dystrophin 3′ UTR is larger than most mRNAs). Translated dystrophin protein is commensurately large, at 427 kDa, and is thus also known as dp427 (dystrophin protein 427 kDa). Dp427 is functionally complex; it has three principal domains: an actin-binding N-terminus, a central “rod domain” containing 24 spectrin-like repeats and a dystroglycan-binding C-terminus (Figure 1D). In skeletal muscle, where dp427 is chiefly expressed, dystrophin is closely associated with the sarcolemma (Figure 1E). Here, it acts as a “bridge”, forming a physical link between the cytoskeletal actin and, through α- and β-dystroglycan, the extracellular matrix environment; this is proposed to act as a “shock absorber”, buffering the membrane stresses associated with muscle fibre contraction. The protein also carries four proline-rich “hinges”, conferring flexibility upon the rod domain, and the spectrin-like repeats themselves are functionally distinct: repeats 11–17 form a secondary actin-binding domain (which can partly substitute for the N-terminal domain [3]); repeats 16–17 further constitute a binding site for neuronal nitric oxide synthase (nNOS), allowing muscle contraction to elicit vasodilatory increases in blood flow through NO signalling [4,5]; repeats 20–23 interact with microtubules (and, consequently, help organise the cytoskeletal microtubule network [6,7]). Finally, the C-terminus is rich with protein–protein interaction domains, recruiting multiple binding partners (both soluble and membrane-bound) in addition to dystroglycan, including the syntrophins, dystrobrevin, sarcoglycans and sarcospan [8]. Collectively, dp427 thus forms a core component of the dystrophin-associated glycoprotein complex (DAGC), a multimeric, multifunctional sarcolemmal assembly that is essential for the maintenance of muscle fibre integrity (Figure 1E).

This picture is further complicated by the fact that dystrophin expression is unconventional: the gene has seven distinct promoters, three of which generate full-length (427 kDa) protein products, and four of which are internal, generating N-terminally truncated proteins that contain distinct subsets of the full-length functional milieu [10]. All these isoforms differ only in their unique first exons, each contributing between 100 and 400 unique bases of sequence (predominantly 5′ UTR), while all remaining downstream sequence is shared (Figure 1C,D). Like dp427, these isoforms are denoted by the molecular weight of the protein product; thus, dp260, dp140, dp116 and dp71, with the three full-length isoforms further delineated by their principal site of expression, to give dp427c (cortical), dp427m (muscle) and dp427p (Purkinje). The major isoform in adults is dp427m, expressed in essentially all skeletal, smooth and cardiac muscle; however, the brain also expresses dystrophin [11,12,13,14], with dp427c (and to a small extent, dp427p and dp427m) found alongside dp140 and dp71, and dp140 is especially abundant within the cerebellum [15,16]. Dp260 is found within the retina [17], and dp116 expression is associated with the Schwann cells of the peripheral nervous system [18]. Dp71 is widely expressed (with the notable exception of skeletal muscle) but is particularly abundant in endo- and epithelial lineages [10,19]. These isoforms are also more widely expressed during embryonic development: dp140 is found within the developing kidney [20], but also within the developing central nervous system (suggesting a role in axonal migration [21]), and we have further shown dystrophin isoform expression during both limb development and tooth maturation [9]. This widespread involvement in fundamental developmental processes suggests an ancient origin for this gene, and this is indeed the case: dystrophin orthologs are found in animal lineages from worms and flies, to fish, birds and humans, and the gene likely predates the arrival of the metazoan kingdom. Interestingly, conservation does not extend to all isoforms: lineage tracing suggests a piecemeal acquisition of shorter isoforms via neofunctionalisation. Dp71 is found in all vertebrates, but dp260 and dp140 are tetrapod-specific (for an excellent overview, see [22]).

### 1.2. Dystrophin and Muscular Dystrophy

The bulk of postnatal dystrophin expression is the dp427m isoform, within skeletal muscle. As discussed above, here, dystrophin holds both structural and signalling roles, forming an essential sarcolemmal buffer against the stresses of muscle contraction. Loss of dp427m leaves muscle fibres vulnerable to contraction-induced injury, and results in the muscle-wasting disease Duchenne muscular dystrophy (DMD). DMD is the single most common fatal monogenic disorder, affecting approximately 1 in 3500–5000 newborn boys every year [23]. The disease currently has no cure and is characterised by repeated cycles of muscle fibre degeneration and compensatory regeneration, with the persistent damage and associated inflammation resulting in the progressive replacement of muscle tissue with fibrotic scar tissue and fat, loss of ambulation and, ultimately, death via cardiac or respiratory failure.

The sheer size of the dystrophin gene renders it susceptible to point mutations through simple probability: this gene represents 0.1% of the entire genome, thus, with ~50–100 de novo mutations per generation [24,25], approximately 1 in 10 individuals carry a new mutation within the dystrophin gene. The gene is also vulnerable to mutational insertions, duplications and deletions (with the latter being common in DMD patients [8]) and, moreover, has mutational “hot-spots”: regions apparently more prone to mutation than others (the major hot-spot being between exons 44 and 53 [26,27]). Mutations that generate a premature termination codon (PTC), either via point mutation (such as in the classic animal model of DMD, the *mdx* mouse [28,29]) or through frameshift following the deletion or duplication of one or more exons, result in essentially no detectable dystrophin protein and, consequently, a DMD phenotype (notably, 7 of the 10 exons within the 44–53 region are vulnerable to frameshift). Mutations causing internal truncations that otherwise preserve the reading frame instead result in a milder condition, Becker muscular dystrophy (BMD). BMD patients often retain muscle function well into adulthood, and some are effectively asymptomatic [30,31]. The extent of internal truncation varies greatly, but even very large deletions (such as exons 13–41) can result in only mild disease [32]. In essence, the dystrophin N- and C-termini appear to be critical for protective function, while much of the internal rod domain is dispensable: this latter observation has driven several potential therapeutic strategies aimed at restoring dystrophin, effectively converting the DMD phenotype to BMD. These therapies take two principal approaches: the exogenous expression of a dystrophin transgene, using “mini-dystrophin” constructs with N- and C-termini but only minimal rod domain (allowing the transgene to be packaged within a viral vector) or reframing via exon “skipping”, eliciting the exclusion of one or more additional exons from endogenous dystrophin to restore the reading frame (either at the transcript level via antisense oligonucleotides or at the genomic level via CRISPR/Cas9-mediated gene editing). The application of this latter approach is contingent on the specific patient mutation, but some exons are more viable targets than others: multiple mutations within the exon 44–53 hot-spot region can be rescued by skipping exon 51, for example. Several of these therapies are now approved for human medicine (or have entered clinical trials), and the accelerating pace of development places increasing emphasis on dystrophin quantification, at both the protein and mRNA levels.

### 1.3. Quantifying Dystrophin Protein

The restoration of dystrophin protein is a primary metric for assessing therapeutic efficacy, as only protein confers resistance to contraction-induced damage (high efficiency of dystrophin correction at gene- or transcript-level is of little value if this does not translate to protein). Quantification of dystrophin protein is comparatively straightforward: there are multiple well-validated antibodies to different dystrophin epitopes, and as dystrophic muscle typically contains no detectable dystrophin protein, western blotting or capillary electrophoresis (or, indeed, immunoaffinity mass spectrometry [33]) can be employed to provide both qualitative and quantitative data [34]. Use of standard curves prepared using known ratios of healthy and dystrophin-negative dystrophic muscle tissue further allows the extent of restoration to be evaluated with precision. It is important to note, however, that such measurements should still be histologically corroborated: fatty/fibrotic replacement will reduce viable muscle tissue (and concomitant apparent treatment efficacy) in a manner not easily discerned from bulk tissue lysates, and furthermore, immunohistochemistry allows the extent of dystrophin restoration to be put into spatial context: an important metric. Establishing that dystrophin protein is (for example) “15% of WT levels” is insufficient, as this could be achieved via 15% restoration in 100% of muscle fibres, 100% dystrophin in only 15% of muscle fibres, or indeed anywhere between these extremes. Even comparatively low levels of dystrophin have been shown to protect myofibres [35,36], thus the modest but global protection offered by the former scenario is likely to prove of more therapeutic benefit than the profound but focal protection of the latter. These approaches also extend to the study of dystrophin isoforms: the very short unique N-terminal sequences mean that isoform-specific antibodies are challenging to generate, however C-terminal antibodies detect all isoforms, which can thus be distinguished electrophoretically on basis of size (for example, to confirm which isoforms are lost in the dystrophic brain [16,34]).

### 1.4. Quantifying Dystrophin mRNA and Transcript Imbalance

Quantification at the mRNA level is commonly employed to determine the efficiency of exon skipping but can also be used to demonstrate the restoration of stable dp427 mRNA via other means (such as gene editing). Study of dystrophin at the transcript level is, however, more challenging than at the protein level. Dystrophin mutations producing a DMD phenotype are typically those generating PTCs, which preclude viable translation of protein product (see above). Presence of a PTC also flags the offending mRNA for prompt degradation via nonsense-mediated decay (NMD), and thus levels of such dystrophin mRNA should ostensibly be low. This is not, however, necessarily the case. Historically, dp427 transcripts were detected by northern blotting [37,38], though the combination of low target abundance and high target molecular weight (alongside small patient cohorts) rendered such approaches challenging: the presence of dp427 mRNA within dystrophic muscle was consequently equivocal. Radiolabelled ISH suggested a substantial nuclear mRNA signal alongside potential enrichment within regenerating myofibres [39], but again such methods were hampered by low target abundance (and the technically demanding nature of radiolabelled ISH). The development of PCR based approaches [40,41,42] permitted more precise detection and quantification of dystrophin transcripts, however reported levels of dystrophin mRNA in dystrophic cells and tissues are still often higher than would be consistent with such NMD-mediated clearance. These findings, combined with the apparent nuclear enrichment of dystrophin mRNA, has led some to propose that mechanisms other than NMD operate on dystrophin transcripts [43]. It is crucial to recognise, however, that the presence of a PTC does not prevent transcription: the pioneer round of translation (the checkpoint for presence of PTCs) occurs after nuclear export, not before [44], and nascent mRNAs (currently being transcribed but not yet completed) are thus not subject to NMD-mediated clearance. For most conventional genes, this distinction is of little consequence: transcription times are sufficiently short such that nascent transcripts represent only a transient, minority population. For mRNAs with lengthy transcription times, however (such as dystrophin), substantial numbers of transcripts might be present in nascent form, detectable via most measures of gene expression (and indeed exclusively within the nucleus) but not necessarily representative of mature mRNA behaviour.

Dystrophin expression also exhibits a phenomenon that has been termed “transcript imbalance”: measured levels of muscle dystrophin mRNA differ according to the region of the transcript targeted. Sequence lying towards the 3′ end of the long dp427 transcript is typically detected at substantially lower levels than that towards the 5′ end (Figure 2). This phenomenon has been noted by us [45] and others [46] and indeed was first recognised by Tennyson and colleagues in the 1990s [1,47]. Of particular relevance, while this 5′–3′ imbalance is more pronounced in dystrophic muscle than in healthy (Figure 2B,C), this phenomenon is, nevertheless, unarguably present within healthy tissue, suggesting it represents a normal facet of dystrophin expression rather than a disease-specific consequence of aberrant transcription.

One proposed explanation for transcript imbalance is premature transcription termination (PTT), where the RNA polymerase complex dissociates and its transcript thus simply fails to be completed. Some mutations in non-coding regions of dystrophin produce a dystrophic phenotype by eliciting PTT [48], and a similar mechanism might also operate under healthy conditions: in essence, the production of dystrophin transcripts might simply have a low success rate. Such proposals are not without merit: while transcription of more conventional genes is typically completed within minutes, the length of the dystrophin gene, and therefore its transcription, might well increase the chances of polymerase dissociation. As noted above, transcription of the entire 2.3 Mb dystrophin gene requires 16 h, suggesting an average transcription rate of ~40 bases a second. This value accords well with those reported for other long genes (40–60 bases per second [49,50]) and implies that this lengthy transcription time cannot be attributed to pausing but instead represents essentially continuous RNA polymerase activity. Spontaneous dissociation of the polymerase complex (and consequent PTT) might be extremely rare, but only a single event is needed to disrupt transcription: the requirement for 16 h of uninterrupted processivity could represent an upper biological limit. Notably, the frequency of stochastic dissociation would increase as a function of transcriptional distance, thus progressive decline in the sequence towards the 3′ end might be an expected outcome.

Another candidate explanation is less intuitive: that transcription is not subject to significant PTT, but that instead mature dystrophin transcripts might have half-lives substantially shorter than the 16 h transcription time. As dystrophin is co-transcriptionally spliced [1], 5′ sequence emerges long before transcript completion and should then persist within the nucleus until polyadenylation and export: levels of this sequence thus reflect both nascent and mature dystrophin mRNA. 3′ sequence is, conversely, not transcribed until relatively late and thus more closely reflects only mature transcripts. If the lifespan of mature transcripts is comparatively brief (i.e., less than 16 h), while the initiation of transcription is concerted, then most dystrophin mRNA will, at steady state, be nascent. Such a model was proposed by Tennyson et al. [47] but could not be empirically confirmed under the technical constraints at the time.

### 1.5. Dystrophin Transcriptional Model

Using a single-transcript multiplex fluorescence in situ hybridisation (RNAscope FISH) approach with probes to the 5′ (exons 2–10), middle (exons 45–51) and 3′ (exons 64–75) regions of the dystrophin dp427 mRNA (Figure 3A), we recently explored this phenomenon at a single-molecule resolution [9,45]. ISH in adult skeletal muscle (Figure 3B) robustly detected all three probes colocalising within the sarcolemma as punctate foci, consistent with triplex-labelling of individual mature dp427m transcripts (Figure 3B inset i, arrowheads). Within myonuclei, where probes would bind to nascent transcripts, a strikingly different labelling behaviour was observed: here, our triplex approach consistently revealed large, intense foci of 5′ probe; slightly smaller, less intense foci of the middle probe; and rare, punctate foci of 3′ probe (Figure 3B insets i and ii). A similar pattern was found within developing myotubes of mouse embryos (Figure 3C). Here, sarcoplasmic mature transcripts appeared more abundant, but nuclear labelling again revealed large 5′ foci, moderate middle probe foci and small, punctate 3′ foci (Figure 3C inset iii and inset schematic). This is consistent with the strong 5′ labelling of myonuclei reported by others [43], and we have shown that these labelling patterns are also found in non-muscle cells expressing full-length dystrophin, such as neurons within embryonic and adult brains of both mice and dogs [9,16], implying that this is canonical behaviour for expression at the dystrophin locus. This phenomenon is consistent with the model proposed by Tennyson et al. [47] and is summarised in Figure 3D: dystrophin transcriptional initiation is robust and continuous, with co-transcriptional splicing ensuring that a viable 5′ probe binding sequence emerges rapidly. Transcription time is long; thus, high numbers of nascent molecules are present within myonuclei at any given time: of these, most will bind the 5′ probe, and approximately half will bind the middle probe, while very few will bind the 3′ probe. Mature transcripts (able to bind all three probes) are rapidly exported, but have modest half-lives and are soon degraded: nascent mRNAs consequently represent the bulk of dystrophin transcripts. Dystrophin mRNA appears to be predominantly nuclear, precisely because most dystrophin mRNAs are still being transcribed.

In further support of this model, in tissues where dystrophin expression is expected to be predominantly composed of shorter isoforms (such as dp140 within the developing nervous system), nuclear labelling is altered accordingly: in the embryonic murine spinal cord (Figure 3E), strong nuclear 5′ foci are found only rarely (associated with the modest, sporadic expression of dp427), however prominent nuclear labelling with the middle probe is retained, indicating nuclei expressing dp140. The transcriptional start site for dp140 lies between exons 44 and 45, and thus nascent mRNAs carrying middle probe sequence (but not exons 2–10 required for the 5′ probe) are present within the nucleus for the ~8 h dp140 transcription time (see model, Figure 3F). Finally, as we have shown previously [9,45], cells expressing the short isoform dp71 label with the 3′ probe only, and exhibit no marked nuclear accumulations: consistent with the short ~1 h transcription time required for this isoform and analogous to the behaviour of more typically sized genes.

This simple model (robust transcriptional initiation, long transcription time and short mature transcript half-life) adequately accounts for these observed phenomena, with the corollary that this same model is sufficient to explain the observed transcript imbalance, obviating any requirement for premature termination. We cannot exclude the possibility that PTT also occurs, however, but this process typically results in the rapid degradation of the incomplete, non-polyadenylated message (up to and including 5′ sequence). Polymerase dissociation should occur stochastically as a function of length and thus be more likely to occur towards the 3′ end: longer transcripts would then indeed be underrepresented. As a consequence of PTT-associated degradation, however, every sequence element of these transcripts (from 5′ to 3′) would be equally underrepresented, essentially resulting in an en bloc reduction in measured levels of dystrophin mRNA, regardless of position. In other words, whether PTT occurs or not, any observed discrepancy in 5′ vs. 3′ sequence can still be wholly accounted for by the combined effects of high transcriptional initiation, long transcription time and short mature transcript half-life.

As we previously noted [45], this transcriptional model is counterintuitive: the implication is that myonuclei continuously initiate expression from the dystrophin locus (our data are consistent with 20–40 nascent mRNAs per nucleus), with each new transcript requiring 16 h of continuous transcription, only for the sarcoplasm to degrade these same transcripts some 4 h after completion. This arrangement, while ostensibly wasteful, is likely of negligible metabolic cost compared to the energetic requirements of muscle activity and interestingly also appears to be conserved among mammals: transcript imbalance is found in humans, mice and dogs [45,46,51]. The enormous size of the dystrophin gene is also largely conserved in other vertebrate lineages, suggesting that this transcriptional model, counterintuitive or not, is likely widespread.

Our proposed explanation is that this arrangement permits circumvention of otherwise absolute biological limits: conventional regulation of expression (supply coupled to demand via control of transcriptional initiation) would be entirely adequate under steady-state conditions and would moreover increase efficiency dramatically (Figure 4A), but here any increase in demand would unavoidably incur a 16 h delay before response and then a concomitant 16 h lag to return to basal levels (Figure 4B,C). Conversely, continuous overproduction matched with post-transcriptional control via degradation (i.e., a supply constitutively in excess of normal demand) results in considerable ongoing waste under steady-state conditions but permits changes in transcript levels (both up and down) to occur over more rapid timescales (Figure 4D–F). Whether cellular demand for dystrophin does, indeed, change markedly under healthy conditions is not presently known, however there are two scenarios under which demand is unarguably high: embryonic myogenesis and muscle repair. In both these conditions, dystrophin levels start at zero but must reach functional levels comparatively rapidly. A system capable of delivering sufficient mRNA to meet these early needs might entail subsequent inefficiency as an inescapable trade-off.

The biochemical and therapeutic ramifications of this transcriptional model merit more comprehensive examination elsewhere: we discuss some aspects below, but the focus of this work is to instead address the profound consequences of the unconventional dystrophin expression programme at the level of basic investigation, from sample preparation to quantitation and data analysis.

### 1.6. Quantitative Measurement of Unconventional Dystrophin Expression

#### 1.6.1. cDNA Synthesis

The generation of copy DNA (cDNA) from RNA via reverse transcription is an essential element of most studies of gene expression. Reverse transcriptases require short oligonucleotide primers as initiators: typically, investigators use oligo dT, random hexamers/nonamers or both. Priming via oligo dT alone allows reverse transcription to commence from the polyA tail (restricting cDNA synthesis to mRNA and avoiding otherwise abundant ribosomal sequences), however polyadenylation occurs only upon transcript completion: nascent mRNAs lack these tails and will consequently be excluded. Furthermore, the low processivity of reverse transcriptase (even highly optimised recombinant enzymes incorporate only ~1500 bases in a single binding event) means that complete reverse transcription of long transcripts requires consecutive cycles of binding and dissociation: 3′ sequence is consequently more readily captured than 5′ sequence in cDNA libraries, and the extent of this 3′ bias increases as a function of mRNA length. For most mRNAs, these nuances are of minimal consequence, but as shown in Figure 5, for dystrophin this distinction is critical: mRNA isolated from dp427-expressing cells will contain a mixture of transcripts at different stages of maturity (Figure 5A–C), only a fraction of which will be polyadenylated. Random priming allows for the capture of representative (albeit fragmented) sequence (Figure 5D); use of oligo dT priming not only excludes all nascent transcripts but also overrepresents 3′ sequence of all mature transcripts (indeed, we have shown that use of oligo dT priming alone can bias prominently against 5′ sequence even in the ~4.5 kb short dystrophin isoform dp71 [52], where nascent transcripts are a minority). In essence, reliance on oligo dT priming gives a representation of dystrophin expression that reflects the exact opposite of biological reality (fortunately, the widespread recognition of 3′ bias means that most investigators recognise the challenges presented by the ~14 kb mature dystrophin mRNA and employ random priming as a matter of course).

#### 1.6.2. Comparing Healthy and Dystrophic Transcripts

This transcriptional model also influences the interpretation of expression under dystrophic conditions. As shown by the phenomenon of transcript imbalance, however, careful choice of target site allows nuanced assessment of transcriptional dynamics. Mutations eliciting a DMD phenotype predominantly introduce premature termination codons (PTCs) into the dystrophin transcript: these will be promptly degraded via nonsense-mediated decay (NMD), but as noted above, this step occurs only upon nuclear export (i.e. after transcript completion; Figure 5F). Consequently, in dystrophic muscle, mature transcripts are greatly reduced as expected, but nascent mRNAs remain (indeed *mdx* muscle myonuclei, both peripheral and centrally located, retain strong 5′ probe foci under ISH [45]). If transcriptional initiation remains otherwise unchanged, RT-qPCR from healthy and dystrophic muscle (Figure 5G(i–iii),H(i–iii)) will thus suggest different extents of NMD depending on the precise region of the transcript selected for measurement. 5′ sequence (exons 1–2) is present in essentially all transcripts (both nascent and mature) and will show only modest fold changes, while 3′ sequence (exons 62–63) is present chiefly in mature transcripts subject to NMD, so here strong reductions will be reported (Figure 5I). Note this also complicates the interpretation of isoform expression (for example, within dystrophic brains): here, first exon sequences are the only means to distinguish isoforms (and thus might not reflect the true effects of NMD), while 3′ sequence is common to all isoforms (and thus will not reflect isoform-specific behaviour). Conversely, changes in transcriptional initiation (in the absence of NMD) will result in en bloc alterations in transcript numbers (Figure 5J–L), leading to more consistent fold changes regardless of the region measured (Figure 5M). These two scenarios are not merely of academic interest: in vivo, both might be present simultaneously, and use of target sites that distinguish predominantly nascent mRNAs (exons 1–2) from predominantly mature mRNAs (exons 62–63) permits the contributions of NMD and transcriptional initiation to be assessed effectively independently. We and others [43,45] have shown that in the *mdx* mouse, reductions in transcriptional initiation do indeed occur alongside NMD (see Figure 2): levels of mature transcripts are markedly lower than in healthy muscle (consistent with degradation), but nascent transcript levels are also reduced, and by a greater factor than can be attributed to NMD (accordingly, under ISH, nuclear foci are prominent but reduced in intensity). Notably, this phenomenon does not appear to be a generalised feature of dystrophic mammals: in the DE50-MD dog model of DMD, loss of mature transcripts to NMD is profound, while transcriptional initiation remains essentially unchanged [51].

#### 1.6.3. Measuring Transcriptional Changes over Time

Temporal studies of gene expression (both in vitro and in vivo) are also necessarily subject to constraints due to this lengthy transcription period. Use of 5′ sequence to study dystrophin transcription during myogenic differentiation would suggest that expression begins earlier (and reaches higher final levels) than would be reported if 3′ sequence were used instead. This also extends to pharmacological modulation of transcriptional behaviour under steady-state conditions: changes in transcriptional initiation will be detected relatively swiftly if analysis is focused on 5′ sequence, but will remain undetectable at the 3′ end for more than half a day. Reversion to canonical expression patterns (such as following pharmacological washout) will similarly exhibit substantial lag, necessitating extreme caution in data interpretation. A 6 h pharmacological blockade of transcriptional initiation followed by a 6 h washout, for example (Figure 6A–E), would generate wildly different findings depending on the specific regions of the transcript selected for analysis. 5′ sequence might largely respond as expected, but 3′ sequence would suggest no response to either treatment or washout over this timeframe. Such a treatment/wash protocol would create a “bubble” of altered transcriptional behaviour moving along the dystrophin locus, and the most profound effects on mature transcripts might not be detected until a full day after treatment began (Figure 6E). Inhibition of post-transcriptional degradation, conversely, would increase measured levels of all sequence regions equally (Figure 6F), essentially applying an en bloc linear transformation (as has been reported following cycloheximide treatment [43]). Expressed as fold change, this increase would first manifest most prominently at the 3′ end where basal levels are low, and these increases would be gradual, commensurate with the slow but continuous production of dp427 mRNAs (5′–3′ differences should also lessen as mature transcripts accumulate, but the prolonged inhibition times necessary for this effect are likely to be incompatible with cell viability).

#### 1.6.4. Quantifying Exon Skipping

Of particular therapeutic interest, this model influences the assessment of exon skipping (targeted exclusion of one or more exons to restore the dystrophin reading frame): if most dystrophin mRNA is nascent and, moreover, co-transcriptionally spliced, many skipped transcripts will be immature at the time of measurement (Figure 7A). Primers spanning the target site (such as exons 22 and 24 for the *mdx* mouse) permit absolute quantification of skipped transcripts regardless of maturity, however expressing this as a fraction of overall dystrophin levels is nontrivial. Quantifying the 5′ sequence to the region of interest (for example, exons 1:2, for “total” dystrophin) underestimates skipping efficiency, as this necessarily includes nascent transcripts that do not yet even contain the target site. Use of a site more 3′ correspondingly overestimates efficiency, as fewer transcripts in total contain such 3′ sequence: indeed, for a skipping site at exon 23, normalising to 3′ sequence could give efficiency estimates in excess of 100%. Sequence closest to the target site gives greatest accuracy (or alternatively sequence unique to unskipped transcripts: measuring across the exon 23:24 splice, for example), however regardless of site, it should be remembered that all measurements are additionally subject to survivorship bias: unskipped mature transcripts are degraded (and, thus, underrepresented), while successfully skipped transcripts are not (Figure 7A(i–iii)).

This picture is further complicated by the dystrophin locus itself: while dp427 mRNA takes ~16 h to complete, individual exons are not spaced equidistantly along the gene (Figure 7B): some thus exhibit more closely matched transcriptional behaviour than others. The first 10 exons, for example, are sparsely distributed across some ~800 kilobases of genomic DNA, while the following 30 exons occupy less than half that. Consequently, specific sequence regions do not emerge in a smooth gradient, but in a more “burst-like” fashion (see Figure 7C), with corresponding consequences for the comparison of skipped/unskipped transcripts (or healthy/dystrophic transcripts). Exon 10 is transcribed almost 6 h after initiation, but the entire following sequence up to exon 41 then emerges in only ~2 h: all exons within this 10–41 region thus represent similar fractions of total dystrophin mRNA (Figure 7D) and will exhibit comparable fold differences between healthy/dystrophic samples (Figure 7E).

The most rigorous approach, consequently, would be to quantify multiple regions independently, in both treated and untreated samples: comparison of early 5′ sequence would assess the influence of treatment on transcriptional initiation, comparison of skipped sequence with adjacent representative sequence would assess skipping efficiency, and comparison of late 3′ sequence would establish the resultant fold enrichment of mature, viable mRNAs.

#### 1.6.5. Dystrophin in the Transcriptomic Era

Next-generation high-throughput transcriptomic approaches (such as RNAseq) are increasingly popular and accessible even to those with modest budgets. As such, the eccentricities of dystrophin expression must also be extended to these techniques, here applying caveats both to sample preparation and data analysis/interpretation. For sample preparation, standard RNAseq pipelines first employ oligo dT column purification (eliminating ubiquitous non-coding RNAs such as ribosomes that might otherwise dominate sequencing). As noted in Figure 5, even this simple step excludes all nascent dystrophin transcripts, a bias then potentially compounded by oligo dT-directed reverse transcription, which favours 3′ sequence. Following fragmentation, ligation and sequencing to FastQ format, analysis first requires the mapping of sequence data to a reference genome to establish the genomic location of each read: these (large) BAM files of aligned reads are then subsequently compared to an annotated feature file to determine which transcripts each genomic location corresponds to, ultimately producing a simple value of “reads per transcript” (or more accurately, “reads per feature”). Post hoc corrections can be applied to normalise for transcript length (longer mRNAs generate more reads), but, conventionally, the specific location of each read within a transcript is not considered relevant to downstream analysis. For a typical RNAseq pipeline, therefore, mRNA isolation eliminates nascent transcripts, reverse transcription biases against 5′ sequence of whatever remains, and downstream sequencing analysis then hides these concerns from the investigator. For dystrophin, in particular, this approach also precludes analysis of isoform-specific expression behaviour: even when isoform variants are present within the feature file, all reads that map to the *Dmd* gene feature are flagged simply as “Dmd”.

Addressing sample preparation concerns is challenging but not impossible: use of ribodepletion instead of oligo dT purification allows retention of nascent transcripts, and similarly (as in Figure 5), the use of random priming (rather than polyA-directed oligo dT priming) allows capture of sequence without 3′ bias (this approach can also be employed to investigate intronic sequence [48]). These modifications cannot be applied post hoc, however, and are thus of no benefit to sequencing data already obtained using conventional polyA purification/priming.

Data analysis is conversely more tractable to post hoc reassessment. Genome feature files (.GFF or .GTF format) store genomic coordinates of each exon and the gene ID of the corresponding spliced transcript, but there is no a priori reason that feature mapping cannot be conducted down to the level of individual exons. This can be done by opening aligned BAM files in a genome browser (such as IGV) and manually counting reads for each exon [53], though this approach is somewhat painstaking. Alternatively, commonly used programs such as Htseq-count can conduct exon-level analysis innately, however this approach necessarily applies to all exons across the entire feature file (generating a vastly excessive dataset), and moreover the nomenclature used to designate individual exons is unwieldy, especially for transcripts with multiple listed variants (such as *Dmd*). For dystrophin-focused investigations, a more practical approach would be advantageous. As gene IDs are simple text fields, and genome feature files can be readily edited (via Excel or text editor), we therefore modified the GRCmm39 mouse genome feature file to add each dystrophin exon as a unique, distinct gene ID (*Dmd_exon3*, etc.) alongside the conventional “Dmd” assignment. This allows mapped reads to be evaluated both overall (total “Dmd”) and at a single-exon resolution (Figure 8A). We further assigned distinct IDs to unique isoform first exons, extending resolution to individual full-length isoforms (dp427c, m, p) and to dp260, dp140, dp116 and dp71 (we note that a similar approach was elegantly employed by Doorenweerd et al. to identify isoform-specific expression patterns in human brain RNAseq datasets [21]).

We first assessed public repository FastQ data taken from a comprehensive transcriptional profiling study of muscle types in healthy mice (generated by Terry et al. [54]), examining three commonly studied pelvic limb muscles: the tibialis anterior (TA), the extensor digitorum longus (EDL) and the soleus (SOL). These datasets were then subjected to a standard analysis pipeline using a standard mouse feature file or our modified version (see methods). This dataset should reflect mature transcripts only: as discussed above, the use of polyA purification excludes nascent transcripts prior to reverse transcription (Figure 5E). Conventional analysis confirmed the authors’ original findings: as expected, myosin heavy chains represented a high percentage (~5%) of total reads but also revealed differences associated with the characteristic roles of each muscle type. Reads per million (RPM) for the very fast IIB myosin heavy chain (*MYH4*) were high (~50,000) in the faster TA and EDL muscles but lower in the slower SOL, with the TA also exhibiting higher levels of the fast IIX isoform (*MYH1*). Conversely, the fast IIA (associated with fast oxidative fibres) and slow Iβ MHCs (*MYH2, 7*) were elevated in SOL but not TA or EDL muscles (Figure 8B). The RPM values for dystrophin (*Dmd*) showed no prominent muscle-specific behaviour, being similar across all samples. Counts for this transcript were also markedly lower (~200 RPM), as expected for a low-abundance transcript.

Reanalysis of these data using our exon-specific dystrophin feature file added substantial context to these findings. The dystrophin 3′ UTR (exon 79) is 2.7kb in length, almost 20% of the mature transcript: one would thus expect reads to exon 79 to be over-represented. Oligo dT primed reverse transcription also biases in favour of 3′ sequence, potentially compounding this over-representation. As shown (Figure 8B, *Dmd* exon 79), this was indeed the case: reads to the (predominantly untranslated) exon 79 represented 30–60% of total *Dmd* reads and were consequently highly comparable to the *Dmd* values obtained via conventional analysis. Reads to all other exons were markedly lower, but 3′ bias was still evident: RPM values at the 3′ end numbered in the hundreds, gradually diminishing to mere tens as exons become more 5′ (Figure 8C). We then adjusted per-exon RPM values for exon length, effectively obtaining an “RPM per base” value for each exon and normalising both for the enormous size of the 3′ UTR and for single reads assigned to multiple exons (see Figure 8A). Expressed in this manner (Figure 8D), the 3′ bias was rendered substantially more obvious and moreover illustrated the highly consistent read counts between muscles. Finally, we plotted these corrected counts against transcript length, using the midpoint positions of each exon within the long dp427 mRNA as X-axis coordinates: as the efficiency of reverse transcription declines essentially as a first-order function of length, plotting in this manner allowed the 3′ bias of oligo dT-primed reverse transcription to be empirically calculated (Figure 8E). For this dataset, we observed a mean gradient of ~−0.0005 log2(RPM).base^−2^, i.e., a two-fold drop in per-base sequence capture for every 2000 bases of distance from the 3′ terminus: again, this gradient was remarkably consistent across muscles, suggesting that this could be used as a dataset-wide correction factor for 3′ bias. Finally, ~95% of isoform-specific first exon counts (Figure 8F) mapped to dp427m, indicating (as expected) that essentially all sequence data represent the muscle isoform of full-length dystrophin. Low levels (<4 counts) of dp116 and dp71 were detected in some samples, consistent with minor contributions from peripheral nervous tissue and vasculature, respectively, but reads to all other isoforms were essentially absent.

Next, we applied this approach to dystrophic muscle, assessing RNAseq data generated by Chemello et al. [55] using the ΔEx51 mouse model of DMD (a gene-edited mouse model that lacks dystrophin exon 51). Conventional analysis of 4-week-old TA muscle samples showed *Dmd* was expressed in healthy samples at levels highly comparable to the mouse muscle data above but was markedly decreased in ΔEx51 samples, as would be expected (the loss of exon 51 causes frameshift, leading to transcript degradation by NMD). Exon-specific analysis again showed that exon 79 reads were over-represented, accounting for ~30% of total (Figure 8G), but also demonstrated that ΔEx51-associated decreases in reads were comparable across the entire length of the transcript (reassuringly, zero reads mapped to the absent exon 51 in ΔEx51 samples; Figure 8H). This consistent decrease suggests these ΔEx51sequences do indeed correspond to mature mRNAs, perhaps captured prior to nuclear export or en route to degradation (note that first exon sequences were nearly exclusively dp427m; Figure 8I). Interestingly, 3′–5′ bias in this instance was consistent with substantially higher efficiency reverse transcription (a two-fold drop every ~7000 bases), potentially explaining the more modest enrichment of the 3′ UTR sequence in this dataset.

Finally, to explore the broader utility of exon-based interpretations, we examined data generated by Schmitt et al. [56], using murine brain samples collected from embryonic day 15.5 (E15.5) to 29 days post birth (P29). Given the developmental expression of dystrophin (especially within the brain), these data should report multiple isoforms, and moreover, sample preparation for this dataset used ribodepletion and random priming, rendering it more representative of both nascent and mature transcription. Conventional analysis readily identified expression of *Dmd*, with expression increasing ~two-fold from E15.5 to P29 (Figure 9A). Exon-specific analysis again demonstrated the strong representation of the exon 79 3′ UTR, but here this exon contributed a smaller fraction than in the studies assessed above (~20%, Figure 9B). Examination of unique first exon reads moreover added considerable nuance to these data, demonstrating that expression of *Dmd* was indeed distributed across multiple isoforms (Figure 9C). Both the cortical full-length isoform dp427c and the short dp71 isoform were robustly detected (and at comparable levels), with expression patterns that broadly mirrored overall *Dmd* expression (though age-associated increases in dp427c were more dramatic than in dp71). Dp140 was the other major contributor, but here expression declined with age, a finding at odds with the behaviour of “Dmd” under conventional analysis, but one that reflects the involvement of this isoform in earlier rather than later neural development. Both muscle and Purkinje full-length isoforms (dp427m, p) were initially essentially absent, but both were present at very low levels by P22–29, while expression of retinal dp260 and peripheral nerve dp116 remained consistent with stochastic noise. Expression of individual exons across the entire dystrophin locus was also markedly different from the patterns revealed above (Figure 9D–I): read counts here were biased in the opposite fashion, exhibiting a clear 5′ enrichment rather than 3′, alongside periodic “spikes” in expression corresponding to distinct initiation events from the dp140 and dp71 promoters (upstream of exons 45 and 63, respectively). The magnitude of these “spikes” differed according to relative isoform abundance within a sample (note that the exon 45-/dp140-associated spike is barely detectable by P29; Figure 9I), which moreover similarly decline from 5′-3′. These data are wholly consistent with the predominantly nascent expression model proposed here: 5′ sequence should be more abundant than 3′ sequence, regardless of isoform, and the contributions of multiple isoforms expressed within a sample (dp427, dp140 and dp71; Figure 9J–L) should thus overlap to create a “saw-tooth” distribution of exon abundance (Figure 9M).

In summary, this brief overview, using established repository-located RNAseq datasets, supports our transcriptional model and, moreover, demonstrates the limitations of polyA-purification and oligo dT priming in the generation of cDNA samples and sequencing libraries. This work focuses only on dystrophin expression, but these findings likely extend to other long transcripts or genes (such as titin and obscurin). We also illustrate the utility of applying a more nuanced exon-level analysis to dystrophin within RNAseq data: this approach offers insights into transcriptional dynamics and isoform expression and can potentially allow broader evaluation of dataset-wide bias.

## 2. Methods

### 2.1. Sample Collection and Preparation

All tissues used here were taken from our tissue archive: no animals were killed specifically for this work. Mouse TA muscle samples were collected postmortem from WT mice (C57Bl/6), mounted on corks with cryoMbed (Bright) in relaxed, longitudinal orientation and snap-frozen under liquid-nitrogen-cooled isopentane before storage at −80 °C. Mouse embryos were collected as previously described [52], fixed in 10% NBF for 24 h and then processed to wax in sagittal orientation for histological sectioning. Embryo used for the work shown here was collected at embryonic day 16.5 (E16.5).

Muscle samples were cryosectioned at −25 °C to 8 μm thickness using an OTF5000 cryostat (Bright) and mounted on glass slides (SuperFrost, VWR, Lutterworth, Leics, UK). Slides were air-dried at −20 °C for 1 h before storage at −80 °C.

Wax-embedded embryos were cooled on ice and sectioned at 4 µm thickness using a microtome (Leica Biocut, Milton Keynes, UK), then floated in a waterbath at 48 °C and mounted on Superfrost slides, as above. Slides were dried at 37 °C overnight and stored at room temperature in sealed containers (with silica gel desiccants, as recommended) until use.

### 2.2. Multiplex FISH: Sample Preparation

Single-transcript multiplex fluorescence in-situ hybridisation was conducted, as described previously [9,16,45], using the RNAScope ISH platform (ACDbio, Aylesbury, UK).

For cryosectioned skeletal muscle, slides were removed from −80 °C storage, immediately placed into cold (4 °C) 10% neutral-buffered formalin and then incubated at 4 °C for 1 h. Slides were dehydrated in graded alcohols, then air-dried and baked at 37 °C for 1 h. Sections were ringed using hydrophobic barrier pen (Immedge, Vector Labs, Peterborough, Cambs, UK) and then treated with RNAscope hydrogen peroxide (15 min) and Protease IV (30 min), per standard RNAscope protocol.

Paraffin-embedded sections were treated according to the RNAscope protocols for FFPE, with target retrieval using the manufacturer’s “alternative method”: slides were immersed slowly in target retrieval buffer (held at a gentle boil) for 15 min, before cooling directly in room-temperature distilled water, followed by ethanol dehydration.

### 2.3. RNAscope Multiplex Assay

Multiplex assays were performed as suggested by the RNAscope multiplex fluorescent reagent kit v2 (ACDbio) protocols, using our mouse dystrophin probe set: Mm-Dmd (452801), Mm-Dmd-O1-C2 (529881-C2) and Mm-Dmd-O2-C3 (561551-C3) (C1, C2 and C3 probes to 5′, 3′ and middle sequence of the dp427 transcript, respectively; see Figure 3A). Nuclei were stained with Hoechst (1/2000 dilution in wash buffer, 5 min), and slides were mounted in Prolong Gold Antifade mounting medium (Thermofisher, Hemel Hempstead, Herts, UK) and allowed to dry overnight (room temperature; protected from light).

Fluorophores were assigned as follows: 5′ probe (C1), TSA-Cy3; middle probe (C3), TSA-Opal520; 3′ probe (C2), TSA-Cy5 (all TSA reagents: Akoya biosciences).

### 2.4. Imaging

Individual images were collected using a DM4000B upright microscope with samples illuminated using an EBQ100 light source and A4, L5, N3 and Y5* filter cubes (Leica Microsystems, Wetzlar, Germany) and captured using an AxioCam MRm monochrome camera controlled through Axiovision software version 4.8.2 (Carl Zeiss Ltd., White Plains, NY, USA). Objectives used were 20× HC PL FLUOTAR PH2 (NA = 0.5).

### 2.5. RNAseq Analysis

All RNAseq analysis used here was conducted within the Galaxy online platform [57] or post hoc using Microsoft Excel. Public repository RNAseq datasets in FastQ format were downloaded and mapped to the GRCm39 mouse genome assembly (RefSeq Acc no. GCF_000001635.27) using HISAT2 [58]. Aligned BAM format files were then mapped to a feature file using htseq-count [59] to determine reads per feature. Use of a custom genome feature file (based on GCF_000001635.27_GRCm39_genomic.gtf), where all dystrophin exons were also assigned unique geneIDs, either according to exon number (exons 2–79) or to isoform (unique first exons), allowed this analysis to evaluate both total Dmd counts and counts per individual exon (to allow reads spanning several exons to be assigned correctly, htseq mapping used “mode: union” and “nonunique: all”). Custom feature file is available on request. Count files were exported to Excel, and raw counts per feature were extracted: counts were converted to reads per million (RPM) to correct for differences in sequencing depth between datasets and (where indicated) then corrected for exon length.

## 3. Discussion

### 3.1. Dystrophin Transcriptional Model

In this manuscript, we present a transcriptional model that accounts for previously reported eccentricities of dystrophin expression observed biochemically [1,43,46,47], and for the behaviour shown here and previously [9,16,45], via multiplex FISH. We further show how dystrophin-focused analysis of next-generation transcriptomic datasets also reveals evidence in support of this model. As noted above, similar models have been proposed historically, but have typically been viewed with caution, or considered to be transcriptional aberrations. As we discuss here, our data suggest that this model is indeed correct, and moreover extends to the expression of some N-terminally truncated dystrophin isoforms. We argue that this model represents normal, functional expression for the dystrophin gene under healthy conditions. As dystrophin expression is restricted to specific cell types, control at the level of transcriptional initiation clearly occurs, but this is essentially on/off Boolean control: with an unavoidable 16 h transcription time, more responsive expression simply cannot be achieved. Continuous overproduction coupled with rapid post-transcriptional degradation thus represents the only way to control expression over meaningful biochemical timescales. Whether such short-term fine control is essential remains an open question. As discussed above, during embryonic development and muscle regeneration (where initial dystrophin levels are zero), overproduction would allow high early translational demands to be met in a timely fashion, with post-transcriptional breakdown rates subsequently increasing to bring levels to steady state. There might however be other, more subtle scenarios under which marked increases in dystrophin supply are required over the shorter term, such as membrane repair. Muscle fibre membranes are subject to considerable stresses even under normal muscle activity, which can cause microtears; the resultant calcium influx initiates a repair cascade that rapidly seals the initial injury and then mediates the remodelling necessary for complete repair [60,61]. Evidence suggests that dystrophin protein is essentially immobile once localised to the sarcolemma [36], implying that the full repair of the sarcolemmal environment requires de novo dystrophin synthesis. At eukaryotic translation rates of ~5 amino acids per second [62], production of a single dystrophin protein requires ~12 min, assuming mRNA is readily available: a modest delay that remains compatible with remodelling-associated repair. A 16 h delay might not be so well-tolerated, and our proposed model would thus represent an effective (if wasteful) solution to this physiological dilemma.

A further question is how transcript degradation is controlled. The extensive 3′ UTR is likely to play a role: studies have shown that the dystrophin 3′ UTR can influence the stability of luciferase constructs [63] (and indeed long 3′ UTRs are known to promote degradation innately [64]). The 3′ UTR of dystrophin is also highly conserved across species, and mutations affecting this UTR can produce both BMD and DMD phenotypes [65], suggesting that its contributions to stability are complex. It is also key to note that while the 3′ UTR (and thus susceptibility to degradation) is shared across dystrophin isoforms, transcription times vary extensively: cells could thus readily achieve higher levels of short dp71 than long dp427, even if both transcripts are subject to the same constant rate of degradation. Degradation could also be modulated as a consequence of disuse: translation factors such as PABP and eIF-4E compete with mediators of mRNA decay [64,66], thus translational activity inherently increases mRNA stability. Given the slow turnover of dystrophin at the protein level (a half-life of weeks to months [67,68]), demand at the mRNA level might be so minimal under healthy conditions that most dp427 transcripts are not translated at all, and are thus promptly degraded as a consequence. This would explain the reported persistence (days to weeks) of therapeutically skipped transcripts in dystrophic muscle [68]: rare, corrected mRNAs would be in high translational demand in a dystrophin-negative context, and therefore continually protected from degradation. This could also explain the domain-restricted behaviour reported previously, where dystrophin protein remains apparently confined to myonuclear territories [36]. Under this model, in healthy muscle, once sufficient dystrophin protein is established within the immediate sarcolemmal territory of a myonucleus, subsequent mRNAs from that nucleus we predominantly be degraded as surplus long before diffusion or trafficking can carry them beyond the domain boundary. This could similarly underpin the marked differences in the pattern of dystrophin restoration reported by Morin et al., depending on therapeutic approach [36]: here, CRISPR/Cas9-mediated genomic correction of *mdx* muscle restored discrete, separate but prominently dystrophin-positive domains, whereas antisense oligonucleotide-mediated transcript correction elicited widespread but more modest restoration. Under this model, genomic correction (via CRISPR) would result in re-establishment only of a local domain arrangement, with these immediately proximal “healthy” levels of dystrophin protein ensuring the rapid turnover of excess corrected mRNAs (with concomitant failure to restore dystrophin protein more distally); conversely, distributed low-level correction at the transcript level (via ASO) would restore only modest levels of dystrophin protein, but more widely, and these “beneficial but sub-normal” levels of protein would not wholly satisfy demand, potentially serving to protect corrected transcripts by virtue of rendering them more translationally active. If dystrophin mRNA stability is indeed strongly influenced by translational activity, “not enough” might prove markedly more effective than expected (this rationale could moreover be applied to dystrophin-positive “revertant fibres” [69,70]: viable transcripts produced by rare aberrant splicing events would be rendered highly stable and thus capable of generating substantial dystrophin protein over time). Factors that interact with the 3′ UTR and mediate stability would be broadly beneficial to multiple therapeutic approaches and thus represent potential therapeutic targets, though we note that within this transcriptional framework, gene therapies utilising mini- or micro-dystrophin constructs (which do not carry the long 3′ UTR) would not be subject to these constraints, and thus transgene mRNAs might be effective even distant from transduced myonuclei.

### 3.2. Dystrophin Expression and RNAseq

We show here how this transcriptional model informs the interpretation of RNAseq datasets, and we illustrate the advantages gained by use of ribodepletion and random priming for reverse transcription (as opposed to polyA purification and oligo dT-directed priming). The analysis approach used here permits more detailed assessments of dystrophin transcription, but we acknowledge that this analysis remains somewhat simplistic. The HTSeq package is primarily intended for differential expression studies, and use of this software for mapping to the level of individual exons is thus not without caveats. A sequence that maps to two exons is counted as a “read” to both exons, even if one is represented by only a single base: our per-base read counts are thus not truly reflective of the exonic read depth, and very short exons might be overcounted purely because such short sequences can be partly present in more reads (for example, the 39-base exon 71 exhibited consistently higher corrected reads than adjacent exons). Similarly, sequences at transcript termini (5′ unique first exons and exon 79) might be undercounted to some extent, as reads to these sequences can only overlap from one end. The substantial length of exon 79 moreover means most reads to this feature map exclusively to this feature, potentially compounding this under-representation (we note that per-base corrected RPM values for exon 79 were consistently lower than adjacent exons; see Figure 8 and Figure 9). Our approach also does not consider alternate splicing: while full-length dystrophin is not held to be alternately spliced in mature skeletal muscle [71], exon 78 is omitted from dp427m at high frequency during embryogenic expression [72,73], and there are multiple splice variants reported for dp71 [74]. Splice events can be identified by eye (using genome viewers), and splice-aware mapping approaches are also available, but these analyses can be challenging, particularly for highly variably spliced transcripts, and, consequently, fall beyond the scope of this manuscript. A further potential source of variability is random priming itself: although necessary to capture representative expression along the entire dystrophin locus, this method assumes that specific hexamer sequences occur with approximately equal frequencies in eukaryotic genomes and that these same hexamers are equally efficient as primers for reverse transcription. Neither assumption is correct [75], and, while this is of only minimal consequence for whole mRNAs, when assessed at the level of individual exons, some regions might well be more readily captured than others. A more nuanced approach would factor in these biases, count mapped reads down to the true individual base level and ideally incorporate identification and enumeration of alternative splicing events: future investigations might address these current limitations (indeed, it would be of considerable interest to repeat this approach comparing polyA/oligo dT and ribodepleted/random primed datasets generated from the same underlying RNA samples).

### 3.3. Dystrophin Transcription: Caveats and Alternative Hypotheses

While the model proposed here is a biophysical inevitability given the size of the dystrophin locus and is, moreover, sufficient to explain the transcriptional eccentricities of dystrophin, this model need not be exclusive: other factors might well contribute. As noted above, we and others [43,45] have shown that dystrophin transcriptional initiation is indeed reduced in dystrophic mouse muscle (which has been ascribed to chromatin remodelling [43]), but this notably does not occur in dystrophic dog muscle [51]. This model also describes only bulk behaviour: our data are consistent with ~20–40 nascent transcripts per myonucleus on average, but this could be achieved by a modest, continuous “trickle” delivery, where nascent transcripts are spaced out along the dystrophin locus, or by a high, infrequent “burst” initiation, where nascent transcripts form a close-packed transcriptional “bubble” moving along the locus. We cannot at present distinguish the two empirically, though we favour the former: under a “burst” system, multiplex ISH should reveal substantial numbers of nuclei positive for 5′ but not middle probe foci, something we do not observe. At the level of transcriptional elongation, aberrant premature transcriptional termination (PTT) might also play a role, though this would necessarily occur on a background already dominated by nascent transcripts (as discussed above). The potential effects of PTT can be approximated by calculating the “fraction of transcripts completed” for a given dissociation constant: with a transcriptional demand of 2.3 million consecutive base incorporation events, a per-base spontaneous dissociation rate of 10^−5^ renders dystrophin transcription effectively non-viable (and, indeed, would significantly impact even substantially shorter genes); at 10^−6^, only ~10% of dp427 transcripts would reach completion; at 10^−7^, completion rates conversely approach 80%. Put simply, there is only a narrow range of per-base dissociation rates over which PTT might be considered relevant. The spontaneous dissociation rate of the eukaryotic RNA polymerase II complex is not known, though studies in yeast imply a high processivity [76]. The error rate (incorporation of incorrect bases) is conversely better studied and is indeed comparatively high (~10^−6^–10^−5^ per base [77,78,79]). The fact these rates are measurable implies that dissociation is likely to occur at a markedly lower frequency than such single-base errors. Furthermore, a significant transcriptional failure rate due to PTT would likely be actively deleterious: a biochemical scenario, whereby transcription cannot reliably be assumed to proceed to completion is one wherein the success of supply/demand logistics is placed at the mercy of stochastics. Under such a model, even constitutive oversupply cannot necessarily be guaranteed to meet demand and is indeed equally likely to result in needless excess. Were dystrophin expression to be limited by transcriptional completion rate, one might well expect strong selective pressure for a shorter, less challenging locus size; the fact that this is not the case across multiple genomes strongly implies that the enormous size of dystrophin is well-tolerated (one might also expect aberrant overexpression in Becker patients with large internal deletions of the *Dmd* gene, but, again, this is not reported). Conversely, an overproduction and post-transcriptional control model is not deleterious: wasteful and counter-intuitive, certainly, but as we discuss here, waste might be a necessary trade-off for a system that is readily capable of meeting both steady-state and variable demand over meaningful biological timescales. One interesting caveat to transcription from such a large locus is that expression is mutually exclusive with replication: 16 h of uninterrupted transcription cannot be accommodated within a typical mammalian S-phase of 8–10 h. As we previously noted [9], the expression of full-length dp427 (and, indeed, dp260 and dp140) is predominantly associated with either muscle or neuronal tissues: both comprised chiefly of post-mitotic cell types. One potential exception is the expression of dp427 in activated satellite cells [80], which occurs prior to asymmetric division. Satellite cells are initially quiescent, however, and satellite cell division itself lags ~15–20 h behind initial damage-associated activation [81,82]: this is consistent with a mitotic delay necessary to transcribe and then translate sufficient dp427. Dp427 is conspicuously not expressed within proliferating myoblasts (though these cells do express the short isoform dp71 [83,84]), and indeed full-length dystrophin expression resumes only once cells have fused to form post-mitotic myotubes.

Finally, we note that dystrophin is not the only long gene: the genes for neurexin-family member *CNTNAP2* and the protein-tyrosine phosphatase *PTPRD* are of comparable size (2–2.3 Mb), and approximately 50 human genes are “very large” (>1 Mb) [85]. Many of the biophysical constraints described here could equally be applied to these other large genes, and indeed the bulk of these 50 genes are associated with post-mitotic muscle or neuronal cell types: it would be of considerable interest to investigate whether these genes also adopt similar transcriptional strategies.

## 4. Conclusions

This primary purpose of this work is to consolidate the eccentricities of dystrophin transcription into a cohesive model and explore the consequences that might result. The data presented remain consistent with this model, though further investigations are merited. This work should hopefully serve as a primer for both molecular biologists and bioinformaticians alike, illustrating the complexities of measuring transcription from such a challenging locus and providing a conceptual framework for the interpretation and analysis of gene expression data derived from both traditional and next-generation approaches.

## Figures and Tables

**Figure 1 biomedicines-11-02082-f001:**
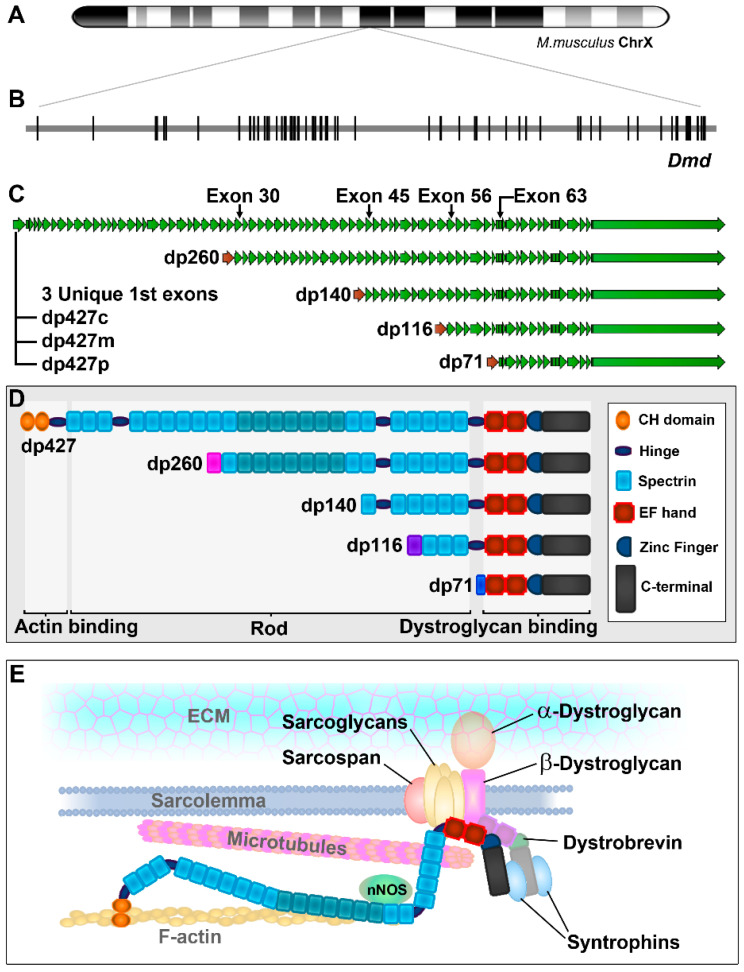
The dystrophin gene. The dystrophin gene is located near the centre of the X chromosome (**A**) and represents ~2% of total X chromosomal sequence. The gene is comprised of 79 canonical exons (**B**), several of which are interspersed with large introns (>100 kb). The gene has seven distinct promoters (**C**), each of which contributes a unique first exon. Three generate full-length dystrophin (dp427c, m and p), while the remaining four are internal, giving rise to N-terminally truncated proteins designated by molecular weight: dp260, dp140, dp116 and dp71. At the protein level (**D**), full-length dystrophin carries an actin-binding N-terminus, a central rod domain of 24 spectrin-like repeats and a C-terminal dystroglycan-binding domain. Repeats 11–17 of the rod domain form a secondary actin-binding domain, and 16–17 bind nNOS. Repeats 20–23 confer microtubule-binding activity. The C-terminal domain also mediates interactions with syntrophins, dystrobrevin, sarcospan and sarcoglycans. Each truncated dystrophin isoform carries a subset of the full dystrophin functional milieu. In skeletal muscle, dystrophin is associated with the sarcolemma (**E**), where it associates with the eponymous dystrophin-associated glycoprotein complex (DAGC), a physical link between actin cytoskeleton and extracellular matrix (chromosomal ideogram adapted from National Center for Biotechnology Information, U.S. National Library of Medicine; other figure elements adapted from Hildyard et al. [9]).

**Figure 2 biomedicines-11-02082-f002:**
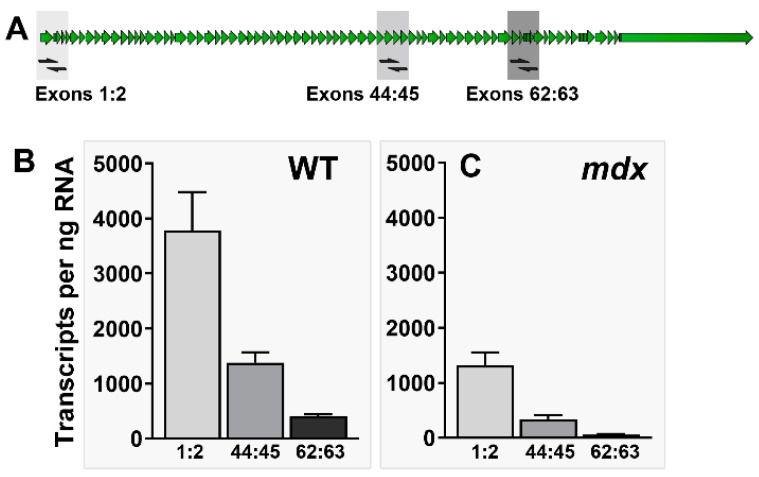
Dystrophin transcript imbalance. Transcript imbalance can be detected using primers to 5′ (exons 1–2), central (exons 44–45) and 3′ (exons 62–63) regions of the 14 kb dp427 mRNA (**A**). Used with cDNA prepared from healthy (**B**) or mdx (**C**) murine skeletal muscle, these primers reveal markedly greater levels of 5′ sequence than 3′. This phenomenon is not dependent on genotype, but measured levels within dystrophic murine muscle show an en bloc reduction, with 3′ sequence being almost absent (figure adapted from Hildyard et al. [45]).

**Figure 3 biomedicines-11-02082-f003:**
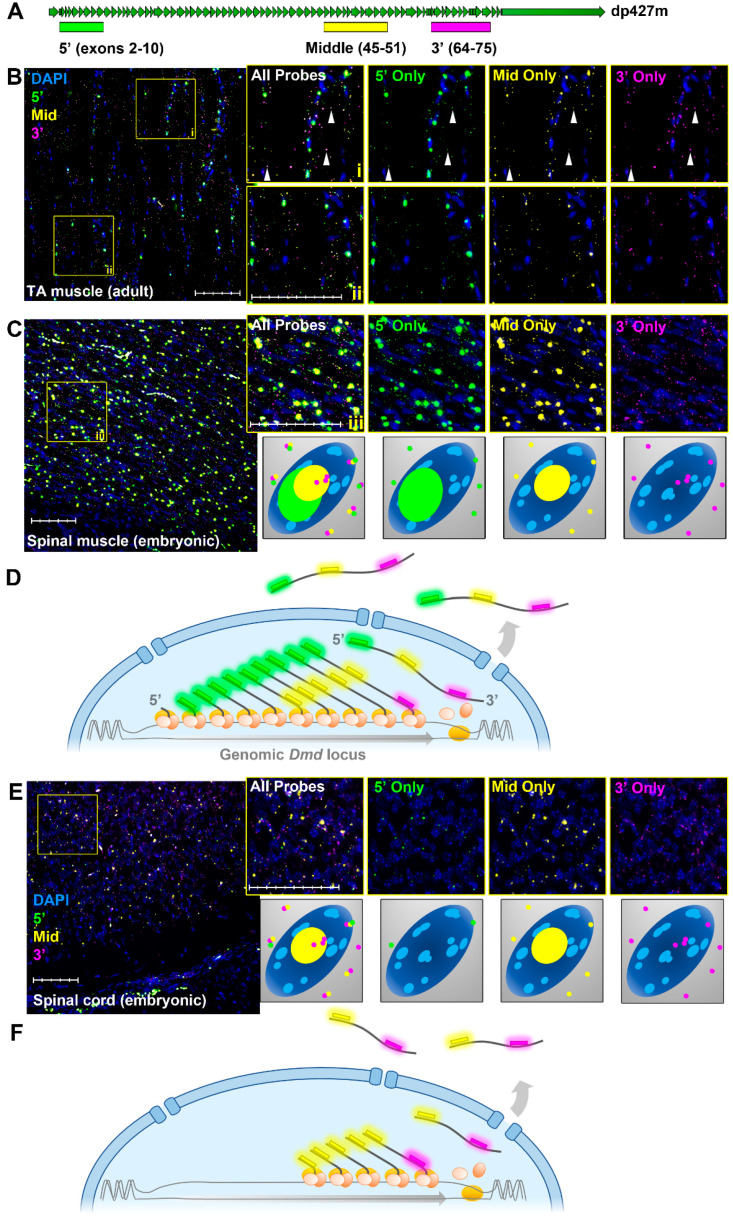
Dystrophin transcription as revealed by multiplex FISH. RNAscope 20-ZZ probes can be designed to the 5′ (exons 2–10), central (exon 45–51) and 3′ (exons 64–75) regions of the dp427 transcript to allow single-transcript multiplex FISH (**A**). Use of these probes in mature skeletal muscle tissue (**B**) or myotubes of developing (E16.5) embryos (**C**) reveals a consistent pattern: sarcoplasmic dp427 transcripts generate punctate foci of all three probes (see arrowheads, magnified insets **i**, **ii**, **iii**), while myonuclei show intense 5′ probe labelling, slightly less intense middle probe foci and minimal, exclusively punctate, labelling with 3′ probe (schematic below (**Ciii**)). A transcriptional model, whereby most dystrophin transcripts are nascent, and mature mRNAs are relatively short-lived, is consistent with this pattern (**D**) and with transcript imbalance shown in Figure 2. Expression of the shorter isoform dp140 within the embryonic (E16.5) spinal cord (**E**) produces prominent nuclear foci of middle probe but not 5′ probe, again consistent with a model whereby transcripts are predominantly nascent (**F**). Scalebars: 200 µm.

**Figure 4 biomedicines-11-02082-f004:**
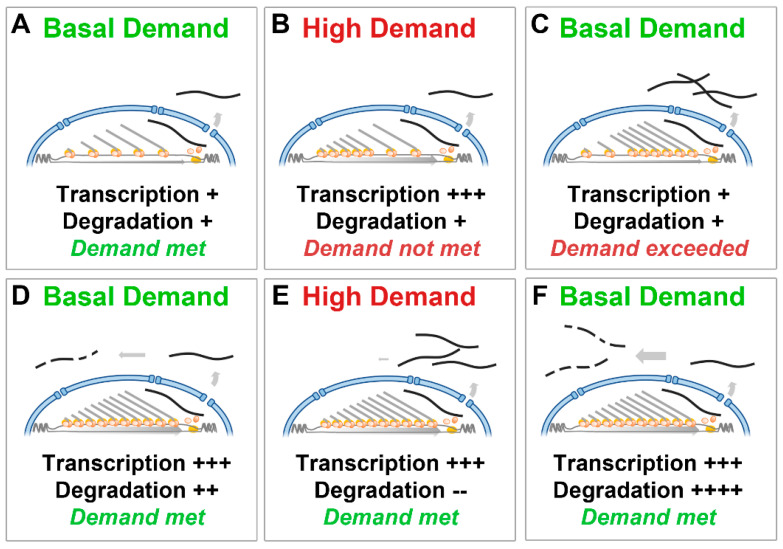
Overproduction with post-transcriptional control circumvents transcriptional delay. A conventional model, where transcriptional initiation matches mRNA demand, is sufficient under steady-state, basal conditions even with a 16 h transcription time (**A**), but increases in demand cannot be met over shorter timescales (**B**), and, similarly, a return to basal demand is also delayed (**C**). A model where transcription is always active and always in excess of demand, with levels controlled post-transcriptionally via degradation (**D**) is constitutively wasteful, but increases in demand (**E**) can be readily met over rapid timescales simply by reducing degradation. Similarly, a return to basal levels can be rapidly elicited by increasing degradation (**F**).

**Figure 5 biomedicines-11-02082-f005:**
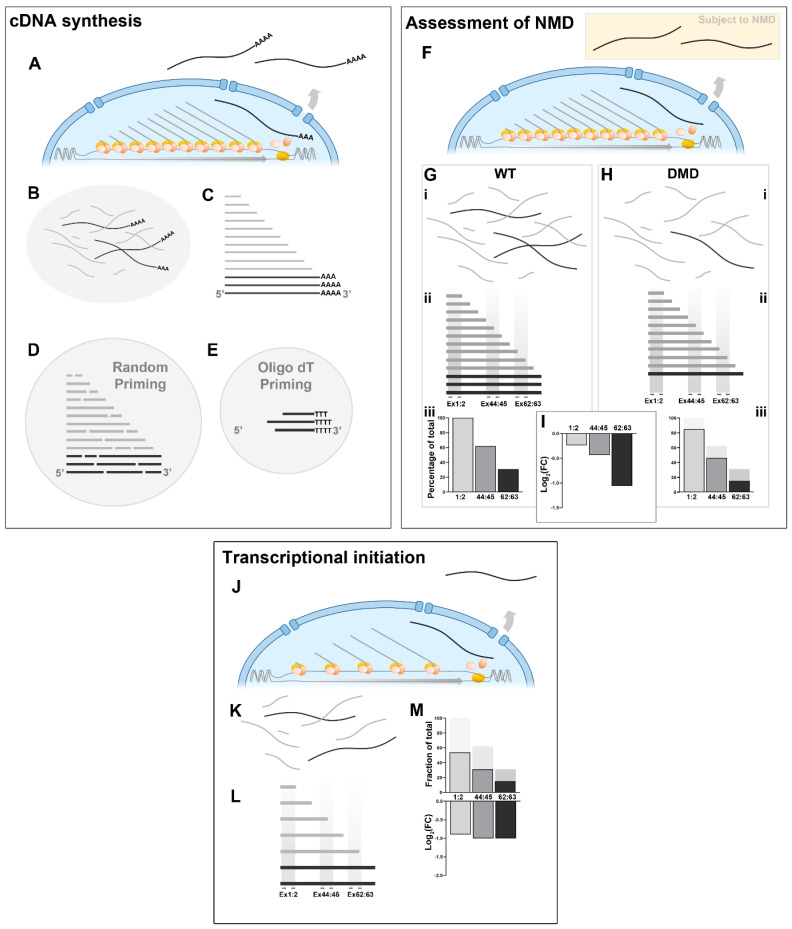
qPCR quantification under the unconventional dystrophin transcriptional model. cDNA synthesis: under this dystrophin transcriptional model, most transcripts are nascent rather than mature (**A**). Following mRNA isolation (**B**), only the full-length fraction of total dystrophin mRNAs carries polyA tails ((**C**), darker lines), while incomplete transcripts do not (lighter lines). cDNA synthesis via random priming (**D**) captures all dystrophin sequence (albeit fragmented), while oligo dT-directed priming (**E**) precludes any capture of nascent sequence and moreover biases towards polyA-adjacent 3′ sequence. Assessment of NMD: this transcriptional model influences assessment of nonsense-mediated decay, as only mature transcripts are subject to degradation (**F**). Assuming transcriptional initiation remains unchanged, both healthy (**G**) and dystrophic (**H**) RNA isolates will contain large numbers of nascent transcripts (**i**), which will be retained following random-primed cDNA synthesis (**ii**), and, consequently, qPCR directed to 5′ sequence will report little or no difference in measured expression, while changes in 3′ sequence will be more profound (**iii**): equivalent WT levels are shown as faint bars (**H**, **iii**). These differences become clearer when expressed as fold changes (**I**). Changes in transcriptional initiation (**J**) will instead produce en bloc reductions in all measured sequences (**K**,**L**), resulting in more consistent fold changes regardless of sequence position (**M**).

**Figure 6 biomedicines-11-02082-f006:**
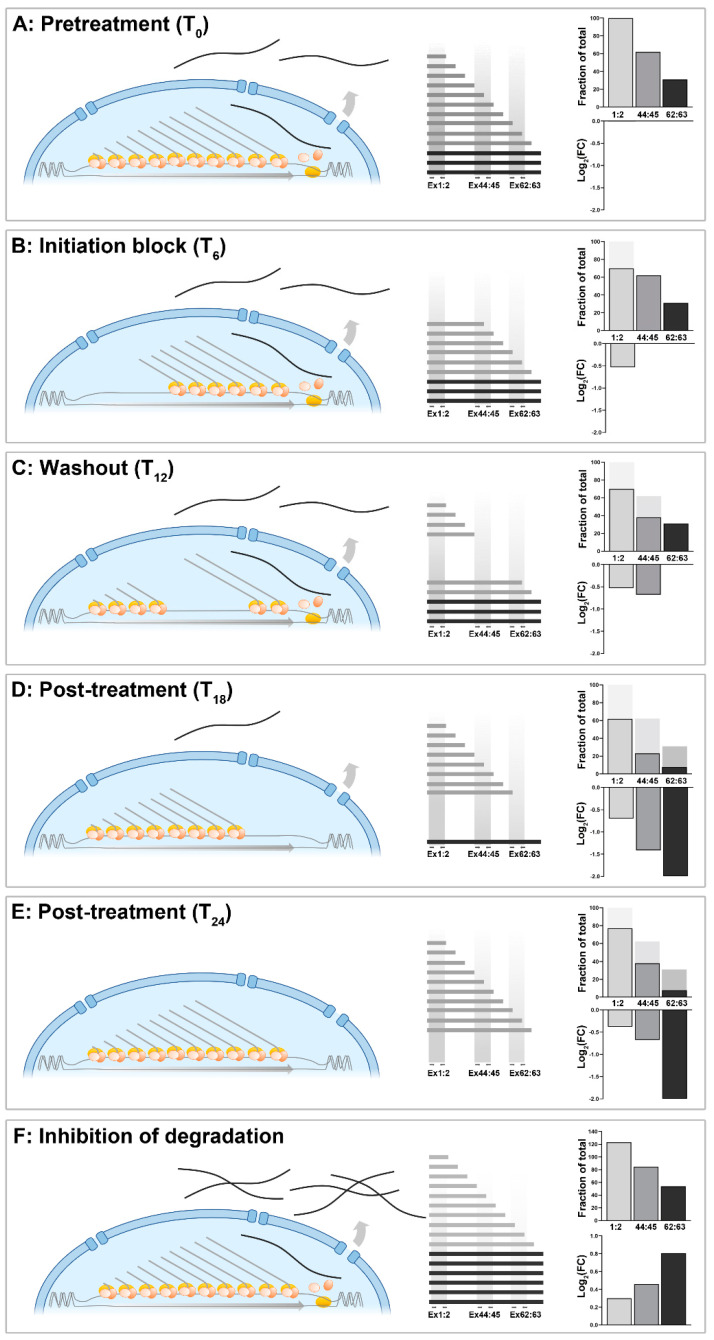
Lengthy transcription times influence responses to pharmacological intervention. Under basal conditions (**A**), most dystrophin transcripts are nascent, and thus measured levels of 5′ sequence are substantially greater than 3′. After 6 h of transcriptional initiation blockade (**B**), only measured levels of 5′ sequence report reductions from basal levels: transcripts initiated prior to blockade persist and are not affected, thus levels of central or 3′ sequence are unchanged. After 6 h of pharmacological washout (**C**), levels of 5′ and central sequence report changes, while 3′ sequence does not: initiation has resumed but a “gap” in the transcriptional procession persists, and transcripts initiated prior to the beginning of the experiment have still not reached completion; 18 h after the start of the experiment (**D**), levels of 5′ sequence remain reduced, while changes in central sequence become more marked, and 3′ sequence levels drop profoundly. A full 24 h after the start of the experiment (**E**), levels of 5′ and central sequence begin to return to basal values, while 3′ sequence remains markedly lower. Blockade of mRNA degradation (**F**) will increase fraction of mature transcripts, leading to en bloc increases in all sequence regions. Fold changes will be more prominent in 3′ sequence, reflecting lower initial levels.

**Figure 7 biomedicines-11-02082-f007:**
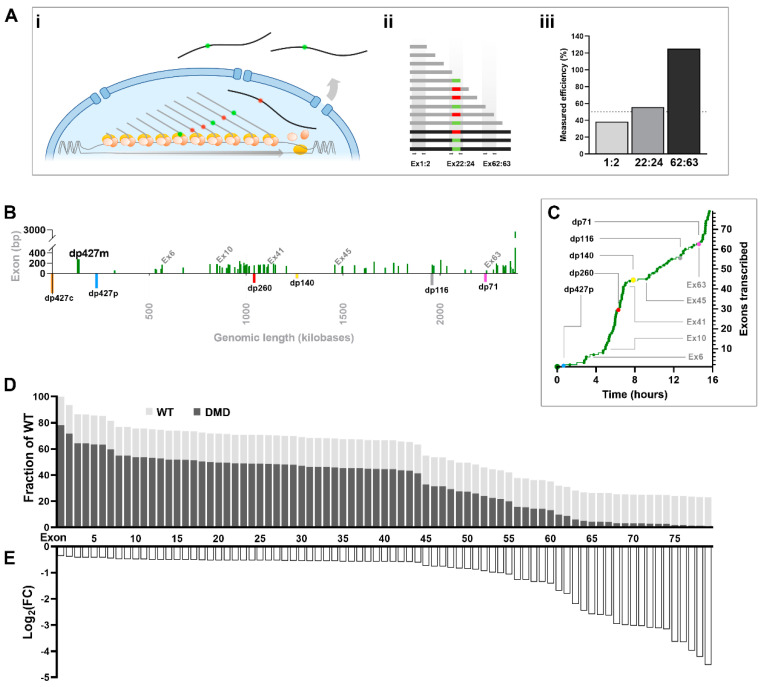
Quantifying exon skipping and accounting for exon distribution. (**A**) Under this transcriptional model, use of antisense oligonucleotides to “skip” exons at the transcriptional level results in nascent transcripts bearing skipped (green regions) or unskipped (red regions) sequence (**i**,**ii**). Only skipped transcripts escape NMD and thus represent the bulk of measured 3′ sequence. This influences measured skipping efficiency (**iii**): only comparison of skipped sequence with sequence close to the skipping site correctly reflects true efficiency (here, 50%: dotted line). Comparison to 5′ sequence underestimates efficiency, while comparison to 3′ sequence will markedly overestimate efficiency. Exons are not distributed equidistantly along the dystrophin locus (**B**), and thus some sequence regions emerge markedly more rapidly than others. X axis indicates bases of genomic sequence; bars represent exon positions. Bar heights represent exon lengths: first exons are indicated. Non-muscle first exons are shown below the X axis for clarity. (**C**) Timeline for sequence transcription: 4 h are required to reach exon 6, while exons 10–41 emerge over ~2 h. All dp71 sequence (unique first exon and exons 63–79) is transcribed similarly rapidly. This renders some exonic regions more susceptible to transcript imbalance than others (**D**,**E**): assuming comparable transcriptional initiation, both healthy (light bars) and dystrophic (dark bars) mRNAs are predominantly nascent, while only mature dystrophic mRNAs are subject to NMD. Dystrophic reductions in 3′ sequence are profound and report dramatic fold changes, while reductions in 5′ sequence might be sufficiently modest to escape detection. The close genomic arrangement of exons 10–41, however, results in comparable (modest) transcript imbalance over this entire region (relative exon abundances are based on the model used throughout this manuscript, where nascent to mature mRNAs are at a ~10:3 ratio).

**Figure 8 biomedicines-11-02082-f008:**
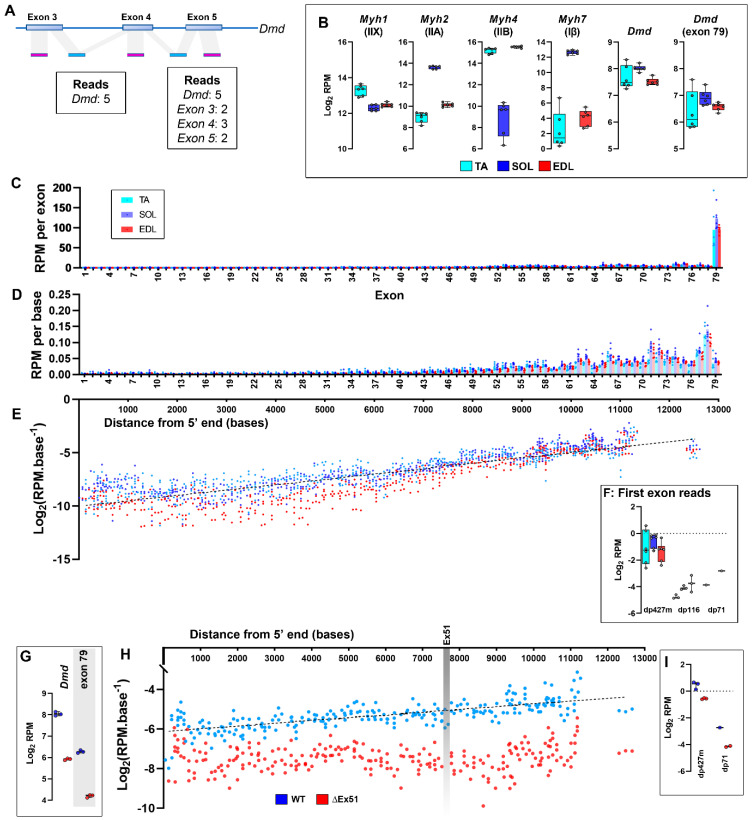
Exon-level analysis of dystrophin expression in healthy and dystrophic muscle. (**A**) Individual sequencing reads (blue/purple) are mapped to genomic features, such as exons of the Dmd locus. Conventionally, all reads to a given gene are summarised regardless of location (left box); however, use of custom feature files allows reads to be mapped on a per-exon basis, giving both overall read counts and counts per exon (right box). Note that reads overlapping multiple exons (blue) count for both. (**B**) Analysis of RNAseq datasets prepared from different healthy murine muscles: tibialis anterior (TA, light blue), soleus (SOL, dark blue) and extensor digitorum longus (EDL, red); all N = 6. Myosin heavy chain expression is consistent with muscle fibre type distribution, with faster MYH genes enriched in faster muscles, while dystrophin expression (Dmd) is comparable regardless of muscle. Counts of Dmd 3′ UTR alone (exon 79) are similar to counts of total Dmd. (**C**) Exon-level reads (reads per million, RPM) along the Dmd transcript show that most reads are to the 3′ UTR. (**D**) Adjusted for exon length (RPM.base^−1^), 3′ bias in read depth is readily apparent and consistent between muscles, and, when plotted against individual exon midpoints along the transcript (**E**), the processivity of reverse transcription can be estimated (−0.0005 log2(RPM).base^−2^, R^2^ = 0.89), corresponding to a 2-fold drop in reads for every 2000 bases from the 3′ end. First exon reads (**F**) are consistent with near-exclusive expression of dp427m. Conventional analysis of healthy (WT, dark blue, N = 3) and dystrophic (ΔEx51, red, N = 3) mouse muscle RNAseq data shows dystrophy-associated loss in Dmd reads (**G**), which exon-level analysis again confirms are chiefly represented by exon 79 sequence. A plot of RPM.base^−1^ against transcript position (**H**) shows loss of Dmd sequence in ΔEx51 muscle is essentially uniform across the entire length of the mRNA (with no reads to exon 51 in dystrophic muscle; see Ex51, shaded region); 3′ bias here is consistent with a 2-fold drop in reads per 7000 bases (−0.00014 log2(RPM).base^−2^, R^2^ = 0.57), and first exon reads (**I**) are again predominantly to dp427m.

**Figure 9 biomedicines-11-02082-f009:**
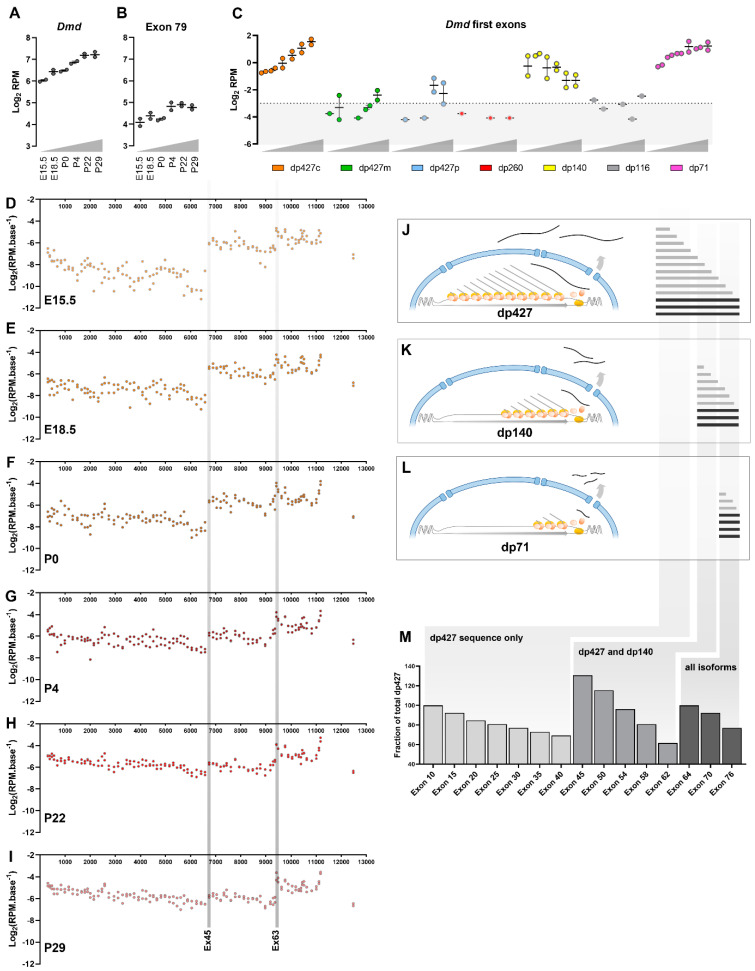
Exon-level analysis of dystrophin expression in embryonic and neonatal brain. (**A**) Conventional analysis of Dmd expression in murine brains collected from embryonic day 15.5 (E15.5) to postnatal day 29 (P29) shows a progressive increase in expression (N = 2 per time point). Exon-level analysis shows that ~20% of this can be attributed to exon 79 sequence alone (**B**), while first exon sequences reveal greater transcriptional complexity (**C**), with expression primarily represented by cortical full-length dystrophin (dp427c), dp140 and dp71. While both dp427c and dp71 show progressive increases in expression with age, expression of dp140 declines. Dashed line and grey box represent read threshold corresponding to stochastic noise (1–2 reads per dataset). Read counts along the transcript (**D**–**I**) show no overt 3′ bias, instead demonstrating 5′ enrichment. Read counts increase markedly at exon 45 and exon 63 (shaded regions). These data demonstrate the advantages of ribodepletion and random priming in generation of RNAseq data for analysis of dystrophin expression and are consistent with a transcriptional model whereby substantial numbers of transcripts are present in nascent form, regardless of isoform: mapping of exonic reads from a mixed sample with expression of dp427 (**J**), dp140 (**K**) and dp71 (**L**) will generate a saw-tooth-like pattern of expression (**M**). Data derived from Schmitt et al. [56].

## Data Availability

All underlying data (and GTF files) used in this manuscript are available from the authors on request.

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
