# Peer review of "When Size Really Matters: The Eccentricities of Dystrophin Transcription and the Hazards of Quantifying mRNA from Very Long Genes"

_biomedicines, 2023, doi:10.3390/biomedicines11072082_

Round 1

Reviewer 1 Report

In this manuscript, the authors have discussed various drawbacks of methods used to detect the dystrophin level. The authors claimed that the long synthesis time of dystrophin with its short half life is responsible for these variation. The manuscript is well written.  I have some minor comments which the authors may address in the manuscript.

1. With the progression of disease in DMD, the muscle fibers are replaced by the fibrotic tissue which may also contribute towards measuring total dystrophin level. The authors may also address this point in the manuscript.

2. The authors didnot discuss the classical Northern blot approach to detect the total level of dystrophin.

3. The authors may also discuss the benefit and drawbacks of different methods of ISH - radioactive vs nonradioactive.

4. Various antibody based method such as IF or western blot have been used to detect dystrophin. The authors may also provide some comments on it to provide a complete picture.

5. The authors have discussed the use of different probes (5', middle, 3') to detect dystrophin. In this scenario, the authors may also discuss the effect of centralized nuclei on the dystrophin level.

Author Response

All of the reviewers comments were extremely helpful, and we thank them for their detailed suggestions. Specific responses are as follows:

  1. With the progression of disease in DMD, the muscle fibers are replaced by the fibrotic tissue which may also contribute towards measuring total dystrophin level. The authors may also address this point in the manuscript.

This is an excellent point, and we have edited the introduction text accordingly (new content in bold):

"It is important to note, however, that such measurements should still be corroborated histologically: fatty/fibrotic replacement will reduce viable muscle tissue (and concomitant apparent treatment efficacy) in a manner not easily discerned from bulk tissue lysates, and furthermore immunohistochemistry allows the extent of dystrophin restoration to be put into spatial context: an important metric."

  1. The authors didnot discuss the classical Northern blot approach to detect the total level of dystrophin.

An entirely reasonable comment: thank you. Detection of dystrophin message via northern blotting historically has presented something of a challenge, both as a consequence of low expression (at least as mature transcripts) and large size (~14kb). We agree that our manuscript directly addresses both these complexities and that mention of northern blotting is thus merited, and we have edited the text accordingly. We have also included a reference to historical ISH-based approaches, as suggested in point 3, below.

Edited introduction text is as follows (new content in bold).

"Presence of a PTC also flags the offending mRNA for prompt degradation via nonsense mediated decay (NMD), and thus levels of such dystrophin mRNA should ostensibly be low. This is not, however, necessarily the case. Historically, dp427 transcripts were detected by northern blotting, though the combination of low target abundance and high target molecular weight (alongside small patient cohorts) rendered such approaches challenging: the presence of dp427 mRNA within dystrophic muscle was consequently equivocal. Radiolabelled ISH suggested substantial nuclear mRNA signal alongside potential enrichment within regenerating myofibres, but again such methods were hampered by low target abundance (and the technically demanding nature of radiolabelled ISH). Development of PCR based approaches permitted more precise detection and quantification of dystrophin transcripts, however reported levels of dystrophin mRNA in dystrophic cells and tissues are still often higher than would be consistent with such NMD-mediated clearance. These findings, combined with the apparent nuclear enrichment of dystrophin mRNA, has led some to propose that mechanisms other than NMD operate on dystrophin transcripts."

  1. The authors may also discuss the benefit and drawbacks of different methods of ISH - radioactive vs nonradioactive.

Again, a helpful comment -thank you. See response above.

  1. Various antibody based method such as IF or western blot have been used to detect dystrophin. The authors may also provide some comments on it to provide a complete picture.

Thank you for this comment. We do address the specifics of protein quantification within our introduction (and indeed discuss the merits of both immunohistochemistry/IF and western blotting) but given that the complexities of dystrophin expression that we discuss within our manuscript are restricted to the transcriptional (rather than translational) level, we feel that further focus on protein-level approaches would distract from our core message.

  1. The authors have discussed the use of different probes (5', middle, 3') to detect dystrophin. In this scenario, the authors may also discuss the effect of centralized nuclei on the dystrophin level.

This is another excellent point, and indeed something of an ongoing puzzle. Persistent central nucleation appears to be largely a mouse-specific phenomenon (myonuclei within regenerated myofibres of dogs and humans ultimately return to a peripheral location), but it is nevertheless very interesting. Our studies have shown that dystrophin transcription is clearly robustly initiated from within centralised nuclei (prominent 5’ foci, smaller but still prominent middle probe foci, comparable with that seen in peripherally located myonuclei), but we do not at present know whether the behaviour of mature transcripts is markedly altered as a consequence. To date our investigations have been restricted to healthy muscle (where centralised nuclei are absent) and dystrophic muscle (where mature transcripts are essentially absent). It would be of considerable interest to investigate the behaviour of mature dystrophin transcripts within healthy muscle subjected to damage (and indeed we hope to conduct such investigations in the near future), and we are delighted that the reviewer clearly finds this as interesting as we do. We have added the following text to the discussion to stress that central nuclei still express dp427, but feel that in-depth discussion of such nuances falls somewhat beyond the scope of this specific manuscript, as they would be largely speculative at this stage.

"(indeed mdx muscle myonuclei, both peripheral and centrally located, retain strong 5’ probe foci under ISH)."

Reviewer 2 Report

The manuscript by Hildyard and Piercy is centered on the technical problems related in transcriptional quantification of very large genes and specifically of the dystrophin gene, and on the significance of the results on the expression of this gene obtained with various methodologies.

The article is well written, interesting and also makes plausible arguments for the presence of very long genes in the mammalian genomes and the related critical issues. In my opinion, the article can certainly be published in Biomedicines.

I have only a technical note: the end of each figure legend is not separated by the following text. I suggest to add at least a space line between figure legend and main text.

Author Response

We thank the reviewer for their positive comments and are pleased they agree with our reasoning. While much of the typesetting/formatting is under the purview of the journal, we are happy to make the spacing between figure legends and body text more obvious (we have added an additional line of spacing accordingly).